# Rainfall Variability and Trend Analysis of Rainfall in West Africa (Senegal, Mauritania, Burkina Faso)

**Zeineddine Nouaceur [1] and Ovidiu Murarescu [2,\*]**

[1] UMR IDÉES CNRS 6266, Rouen University, 76821 Mont Saint Aignan CEDEX, France; zeineddine.nouaceur@univ-rouen.fr

[2] Department of Geography, Valahia University, 130001 Târgovişte, Romania

\* Correspondence: ovidiu.murarescu@valahia.ro

**Abstract:** This study concerns the West African Sahel. The Sahelian climate is characterized by a long dry season and a rainy season which starts in June and ends in September–October. This latter season is associated with the process of oceanic moisture transfer to the mainland (the West African Monsoon). This movement is governed by an overall moving of the meteorological equator and its low-pressure corridor (Intertropical Convergence Zone, ITCZ) towards the north, under the effect of the attraction of the Saharan thermal depressions and a greater vigor of the anticyclonic nuclei. This study was conducted on 27 Sahelian climatic stations in three countries (Burkina Faso, Mauritania, and Senegal). The method used to determine the modes of this variability and the trends of rainfall is the chronological graphic method of information processing (MGCTI) of the "Bertin Matrix" and continuous wavelets transform (CWT). Results show a rain resumption observed in the recent years over the Sahelian region and a convincing link with the surface temperature of the Atlantic Ocean.

**Keywords:** West Africa; climate change; rainfall variability

## 1. Introduction

If, on a global scale, the rise in temperatures is a certainty, the evolution of global rainfall is much more contrasting, as it is subject to a strong spatiotemporal variability [1].

Increased power of evaporation will lead to a greater availability of water vapors. An increase in the quantity of the humidity in the lower layers of the atmosphere might be the cause of an intensification of rainfall in a torrential form [2]. Using a climate simulation model, it was proved that a 22% rise in air humidity may be brought towards continents by maritime flows [3]. The results of various studies regarding rainfall evolution also show that climate changes have entailed an intensification of precipitation and a repetition of extreme events [4–8].

Given this climate change, a likely increase in extreme events, particularly floods, is to be expected. According to the World Meteorological Organization, floods are the most frequent extreme phenomenon that occurred in the 2001–2010 decade [9]. This phenomenon affected several regions of the World, producing hundreds or even thousands of victims. In 2016, almost 23.5 million people were dislocated because of natural disasters related to extreme meteorological events—mainly storms and floods in the Asia-Pacific region [10]. In Europe, a significant change in the flood calendar due to current climate changes [11] was highlighted. Some researchers have pointed out increasing trends of extreme rainfall in more than 8326 weather stations worldwide [12]. Furthermore, they have been able to prove a significant statistical association between the average temperature and the meridian variation (the highest sensitivity occurs in the tropics and at high elevations, whereas the highest uncertainty is near the Equator, owing to a limited number of sufficiently long recordings on precipitation). The Sahelian climate is characterized by a long dry season and a rainy season which starts in June and ends in

September–October. The latter season is associated with the oceanic moisture transfer to the mainland (the West African monsoon). This movement is governed by an overall translation of the meteorological Equator and its low-pressure corridor (Intertropical Convergence Zone—ITCZ) towards the north, under the effect of the attraction of the Saharan thermal depressions and the intensification of the anticyclonic nuclei in the southern hemisphere in winter. The Sahelian climate is subject to a very high rainfall variability. In the 1970s–1980s, the scientific world mobilized to conduct studies on climatic droughts, which was one of the most inter-decade powerful signals ever observed on the planet. Researchers are currently focusing on the African monsoon in order to elucidate the complexity of mechanisms related to its spatiotemporal variability. The international program, African Monsoon Multidisciplinary Analysis (AMMA) illustrates this approach [13].

The Eastern Equatorial Atlantic is the region of the tropical Atlantic basin where the seasonal cycle of ocean surface temperatures is most marked (a drop of 5 to 7 °C is observed in spring). This anomaly called "cold water tongue" [14] appears around 10° W and extends to the area south of the equator between the African coasts and 20° W. Under the effect of this contrast, the southeast trade winds cross the equator, pivot in a southwest wind, and penetrate the continent where they form the monsoon flow in the lower layers of the atmosphere.

It was the presence of abnormally cold waters in the North Atlantic and abnormally warm waters in the South tropical Atlantic that was first associated with Sahelian rainfall deficits, leading to a reduced rise in rainfall to the north [15]. Another research suggests that this Atlantic dipole structure was part of a global inter-hemispheric structure linked to the strongest warming of the tropical ocean recorded in the last century [16].

Regarding the research of the role of sea surface temperature (SST), Tropical Southern Atlantic (TSA), and Tropical Northern Atlantic (TNA), this is frequently highlighted as a source for decadal variability of rainfall in West Africa [17,18]. Atlantic multidecadal variability has been given much attention, particularly regarding the interhemispheric pattern of SST and Atlantic multidecadal oscillation (AMO) [19–24].

In terms of the surface temperature of the oceans, predicting the onset of this climatic phenomenon and knowing its intra-seasonal and annual variability is crucial for local populations. It is known this region is vulnerable to drought on a large scale, because some of its economy is based on an agricultural system dependent on precipitation. The Sahelian economy is in its infancy with a human development index (HDI, this index varies from the highest value 0.954 for Norway to the lowest value 0.354 for Niger) calculated for all countries in the studied region of 0.43 (Mauritania 0.52; Senegal 0.50; Mali and Burkina Faso 0.42; Chad 0.40; and Niger 0.35).

## 2. Data and Working Methods

This study relies on data from 27 Sahelian weather stations in three countries (Burkina Faso, Mauritania, and Senegal). Annual data for the period between 1947 and 2014 has been used for the chronological graphic method of information processing (MGCTI) of "Bertin matrix" (this choice was prompted by data availability—Figure 1). The mean monthly values were available only for stations in Senegal and Burkina Faso. Available monthly rainfall data were extracted between 1948 and 2017 for each country for the continuous wavelets transform (CWT) analysis. The data used have not been interpolated but come from measurements from observation stations that appear on the map in Figure 1.

The data were collected from the weather services in Senegal (ANACIM) (http://www.anacim.sn/meteorologie/), Mauritania (http://www.onm.mr/). Rainfall data from Burkina Faso were collected from the KNMI Climate Explorer (http://climexp.knmi.nl/selectstation.cgi?id=someone @somewhere). Some missing data were found on "TuTiempo.net" (https://fr.tutiempo.net/climat/afrique.html). This site uses the National Climatic Data Center's Global Database (NCDC—https://www.ncdc.noaa.gov/data-access/quick-links).

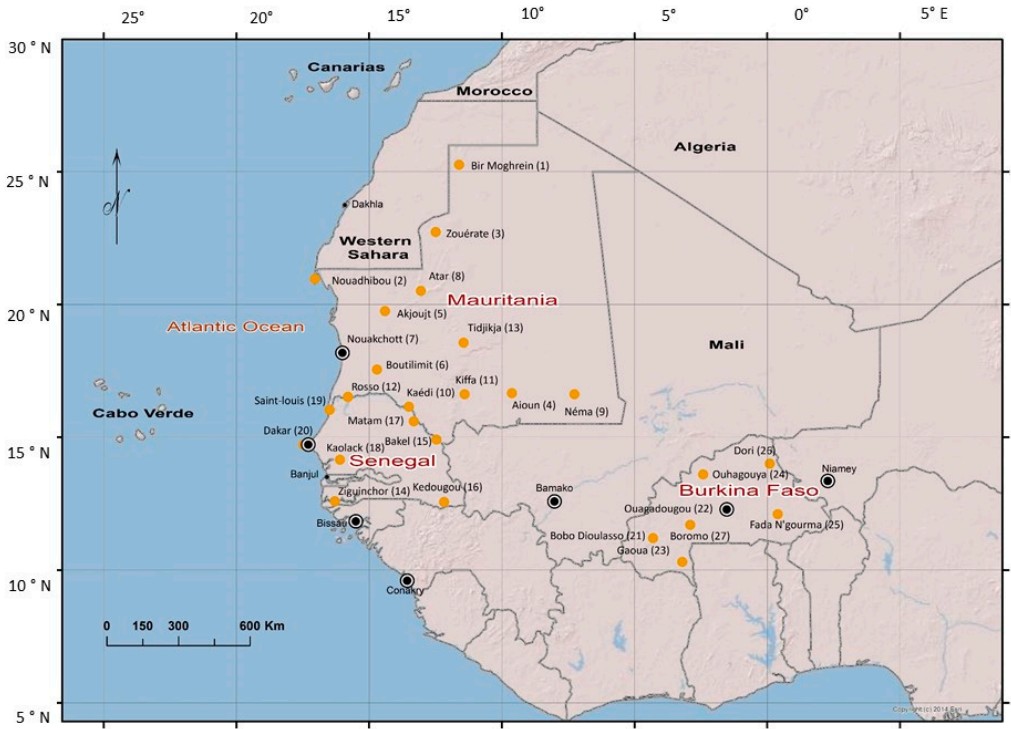

**Figure 1.** Geographical position of the 27 weather stations cited. The number between the brackets indicates the position of stations in the chronological graphic method of information processing (MGCTI) of "Bertin matrix" type (Figure 2).

Three regionalized indices of surface temperatures of the North Atlantic and South Atlantic Tropical, freely accessible on the site (https://www.esrl.noaa.gov/psd/data/climateindices), were used: The AMO index (25–60° N, 7–75° W), the TNA anomaly of the average of the monthly SST from 5.5° N to 23.5° N and 15° W to 57.5° W), and the TSA anomaly of the average of the monthly SST from Eq-20° S and 10° E–30° W).

The analysis of rainfall variability employs two working methods:

- The chronological graphic method of information processing of "Bertin matrix" type. The MGCTI is an analytical method based on a statistical analysis and on a graphical representation of results. This method was used for the first time in 2013 [25,26]. The MGCTI is developed to facilitate the interpretation of the statistical results for the Mediterranean rainfall analysis, due to the high variability affecting this parameter. The Bertin matrix was introduced to harmonize and consolidate information after the statistical treatment. The Bertin matrix is a manual and visual method of classification of information based on data. This matrix is used to group the data that have similarities according to all the criteria studied. It provides a simple and effective way to establish a multivariate typology based on observations of the user. This provides the concordant results (even rainfall character for all stations studied in the same year) but also identifies conflicting information (different characters between stations for the same year). The MGCTI and its graphic representation allow a chronological reading and a spatial analysis of the phenomenon. This method has been successfully tested in many North African regions [27–30], and in the Sahel [31]. A comparative study with the Standard Precipitation Index (SPI) method for detecting climate drought was conducted in 2015 [32]. This study showed the simplicity and clarity of the results obtained with the MGCTI method. One of the aims of this article is to show the trend of rainfall over nearly half a century and to detect the date of changes of cycles.

The First Stage. An annual data precipitation (cumulative rainfall over a calendar year) hierarchy in terms of limit values (Q1, Q2, Median, Q3, and Q4) is done for all stations and for the entire series

(Table 1). Depending on data position in relation to limit values, the years are considered as very dry, dry, normal, rainy, very rainy (Table 2):

(i)   very dry, below the first quintile;
(ii)  dry, between the first and the second quintile;
(iii) normal with trends towards drought, between the second quintile and the third quintile;
(iv)  rainy, between the third and the fourth quintile;
(v)   very rainy, above the fourth quintile.

**Table 1.** Distribution and hierarchization of annual rainfall according to the quintiles.

| Mauritania | Q1 | Q2 | ME | Q3 | Q4 |
|---|---|---|---|---|---|
| Aioun | 146 | 209.4 | 221.1 | 256.6 | 325.6 |
| Akjoujt | 32.6 | 54.7 | 72.8 | 84.8 | 137.5 |
| Atar | 38.9 | 68.3 | 76.9 | 90.4 | 127.8 |
| Bir Moghrein | 8.8 | 21.7 | 27 | 37.1 | 70.5 |
| Boutilimit | 96 | 135.2 | 159.5 | 170.8 | 227.4 |
| Kaédi | 218.8 | 251.1 | 288.9 | 319.9 | 415 |
| Kiffa | 176.1 | 264.1 | 281 | 319.2 | 422.9 |
| Néma | 164.5 | 220.4 | 231.5 | 258.2 | 331.5 |
| Nouadhibou | 4.9 | 12.5 | 19.9 | 24.3 | 45.6 |
| Nouakchott | 45.4 | 81.7 | 95.4 | 119 | 189.7 |
| Rosso | 163.3 | 239.2 | 258.5 | 297.3 | 338.9 |
| Tidjikja | 53 | 95.5 | 110.2 | 130.2 | 171.7 |
| Zouérate | 21.3 | 38 | 47.2 | 56.7 | 85 |
| **Senegal** | | | | | |
| Bakel | 397.1 | 468.7 | 503.4 | 542.3 | 670 |
| Dakar | 274.3 | 382.8 | 421.6 | 477.2 | 624.3 |
| Ziguinchor | 1105.3 | 1281.5 | 1383.2 | 1531.4 | 1677.3 |
| Saint Louis | 188.7 | 240.6 | 278.9 | 300 | 373.5 |
| Kedougou | 1062 | 1154.5 | 1186.3 | 1264 | 1375.8 |
| Kaolack | 510.5 | 590.2 | 636.6 | 716.5 | 849 |
| Matam | 292.7 | 368.9 | 412.2 | 452.4 | 522.5 |
| **Burkina Faso** | | | | | |
| Bobo Dioulas | 888.8 | 972 | 1037.7 | 1085 | 1246 |
| Boromo | 773 | 884 | 930 | 963 | 1051 |
| Dori | 397 | 455 | 476 | 531 | 625 |
| Fada N'gourma | 699.5 | 789 | 824.5 | 907 | 999.5 |
| Gaoua | 954 | 1024 | 1059 | 1096 | 1207 |
| Ouagadougou | 675 | 738.87 | 765 | 803 | 928 |
| Ouhagouya | 536 | 612 | 649 | 709 | 767 |

The Second Stage. A recoding of values is made by means of a range of colors (the color varying in terms of the annual cumulative rainfall position in relation to limit values). This first processing is followed by a reordering procedure (permutations of columns) in order to get a ranking that allows

the visualization of a homogenous colored structure (Bertin matrix) (Figure 2). This procedure allows for the visualization of the climate parameter evolution in terms of two dimensions (time and space).

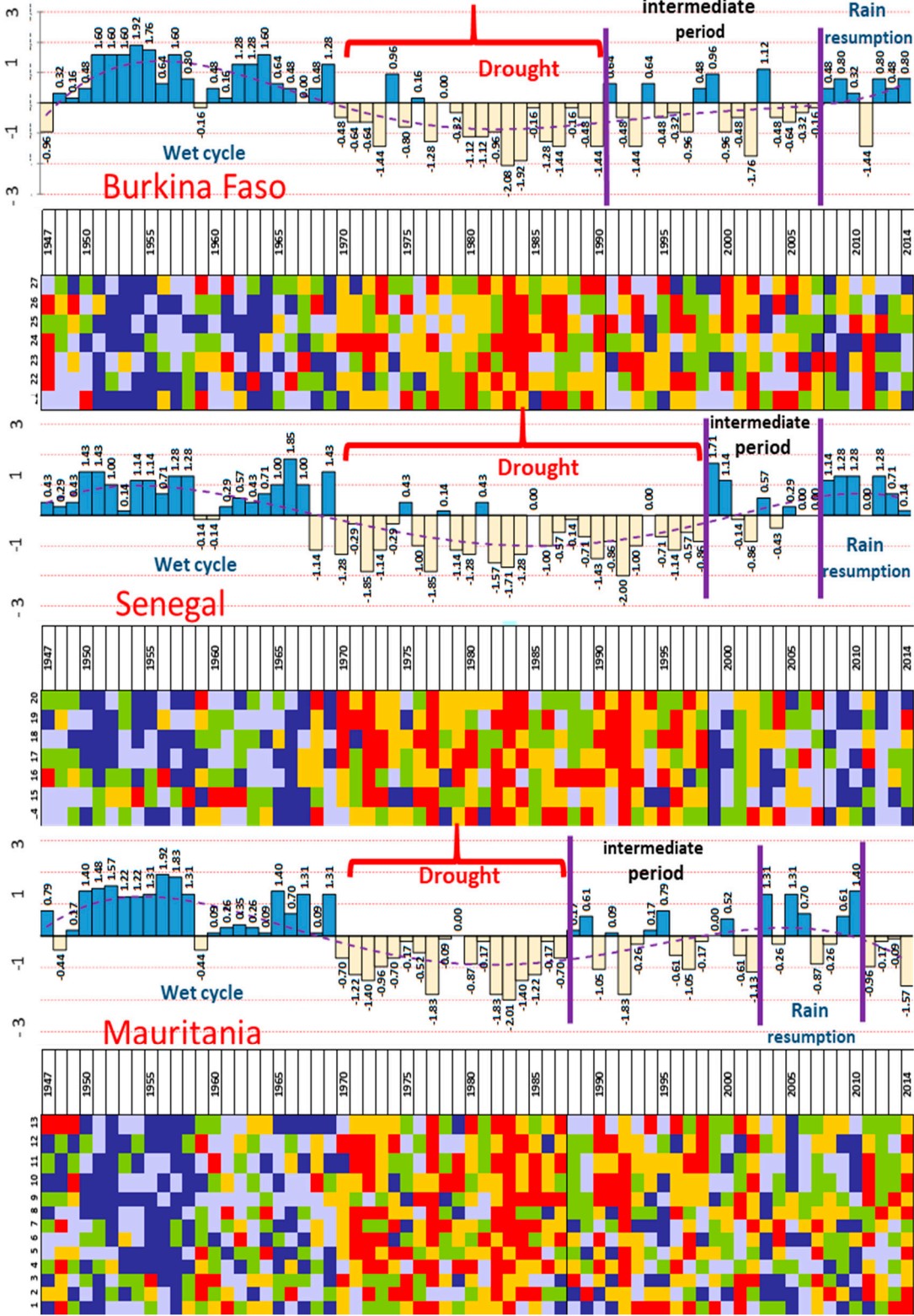

**Figure 2.** Chronological graphical matrix of data processing applied to rainfall (1947–2014).

**Table 2.** Trend of annual rainfall according to the quintiles.

| Thresholds | Q1 | Q2 | Median | Q3 | Q4 |
|---|---|---|---|---|---|
| Distribution of values (%) | 0–Q1 (0–20%) | Q1–Q2 (20–40%) | Q2–Median (40–50%) Median –Q3 (50–60%) | Q3–Q4 (60–80%) | >Q4 (80–100%) |
| Annual rainfall | Very dry | Dry | Median | Wet | Very wet |
| Trend at station | | | | | |
| Regional trend | | | Not expressed | | |

The Third Stage. To determine the typical breaks and periods, a second procedure is conducted. It consists in assigning a number ranging from one (very dry year) to five (very wet year) according to the already determined features assigned to each year. The sum of numbers of all stations for each year is centered and reduced, thus getting a regional index (*RI*) varying from +∞ for a very wet year to −∞ for a very dry year. The "*RI*" is calculated as follows:

$$RI = (X_i - X)/S \tag{1}$$

where $X_i$ is yearly value. $X$ is the series average, and $S$ is standard deviation. The projection of the result on a graph allows for the visualization of the evolution of the phenomenon on a regional scale in a first stage and. in a second stage for the determination of data on breaks and trend change.

- The continuous wavelets analyses allow a temporal location of the variability of a given signal. It breaks down the signal, both, in time and in frequency, which can correctly describe these hydrological or climatic fluctuations, periodic or not [33] introducing the transformation into wavelets that, unlike the Fourier's transformation, breaks down the signal into a sum of finite-sized functions located over time for each frequency detected in the signal. For this, a mother wavelet is dissociated into girl waves to find the given frequency and is then translatable to analyze the neighboring frequencies. So, these analyses were developed to compensate for the disadvantages of conventional Fourier analysis. The girl wavelets have the result of the decomposition of the reference wavelet (mother wavelet). Each wavelet has a finite length (a ladder) and is highly localized over time. The mother wave has two parameters for time-frequency exploration: A scale $a$ and a time location $b$:

$$\psi_{a,b}(t) = \frac{1}{\sqrt{a}}\psi\left(\frac{t-b}{a}\right) \tag{2}$$

The setting in scales and the translation of the girl wavelets allow the detection of the different frequencies that make up the signal. In addition, these frequency components can be detected and studied over time, allowing for a better description of non-stationary processes [34]. The continuous wavelet of an *S(t)* signal produces a local wave spectrum, as defined by (Equation (2)):

$$S(a,b) = \int_{-\infty}^{+\infty} s(t) \times \frac{1}{\sqrt{a}} \times \psi\left(\frac{t-b}{a}\right) \times dt \tag{3}$$

The convolution of the filtered monthly signal (Rainfall, AMO, TNA, and TSA) by a non-orthogonal wavelet basis was applied to define the continuous wavelet time-scale spectrum, which is able to identify the spectral components assigned to the dominant mode of variability of the total signal.

The rainfall signal was analyzed using CWT to identify the dominant modes of variability characterizing the Sahelian rainfall (Figure 3). Furthermore, the convolution of this signal with the wavelet basis generates a contour diagram having three variables: (1) the time graduation on the x-axis, (2) the time scale of wavelet on the y-axis, and (3) the power or variance of variability on the z-axis

which can also can be explained by the correlation degrees between the signal and the wavelet basis. Such representation illustrates eventual changes in the variance based on the non-stationary signals. Distribution of the power (expressed as normalized decibels, i.e., maximum power = 0 bd for each scale) in the wavelet contour diagram is assigned by a variation of color from dark blue, that represents the low power (variance), to dark red, which represents the increasing power observed.

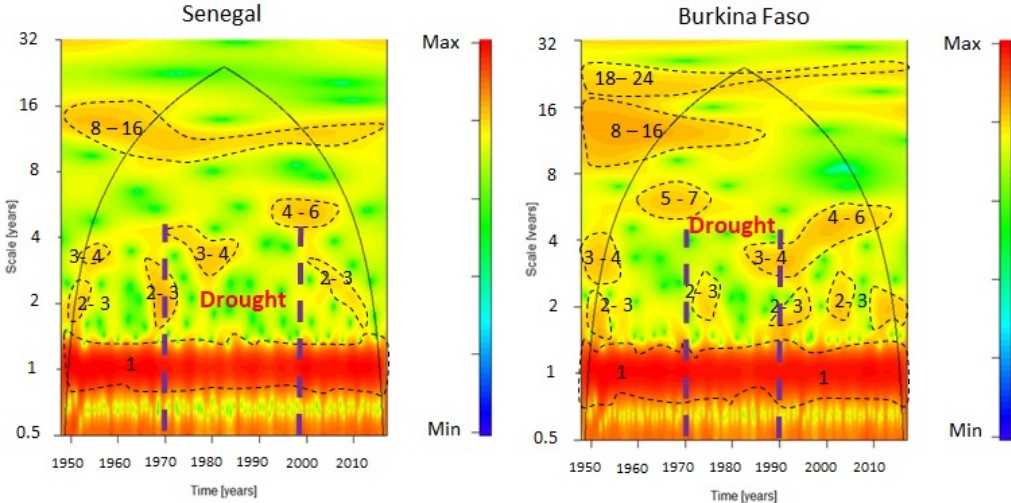

**Figure 3.** Continuous wavelet of precipitation regime in Senegal and Burkina Faso.

Then, correlation between the index of climate AMO, TNA, TSA, and rainfall signal was computed by the coherence diagram (WCO) for different modes of variability.

To compare the time series with each other, cross-correlation analysis is used.

By analogy to the spectra crossed by Fourier transform, the spectrum in crossed wavelets is a method which makes it possible to evaluate the correlation between two signals according to the various scales (frequencies) during time [35,36].

The spectrum by crossed wavelets $W_{xy}(a, \tau)$ between two signals x(t) and y(t) is calculated according to the equation below, where $C_X(a, \tau)$ and $C^*_Y(a, \tau)$ are the wavelet coefficient of the continuous signal x(t) and the conjugate of the wavelet coefficient of y (t), respectively:

$$W_{xy}(a, \tau) = c_x(a, \tau) \times C^*_y(a, \tau) \tag{4}$$

Continuous wavelet coherence can be defined as the estimate of the temporal evolution of linearity and of the relationship between two signals on a given scale [34,37]. Wavelet consistency is calculated using smoothed wavelet spectra of the $SW_{xx}(a, \tau)$ and $SW_{yy}(a, \tau)$ series and a smoothed crossed wavelet spectrum $SW_{xy}(a, \tau)$ [38].

$$WC(a,\ T) = \frac{|SW_{xy}(a, \tau)|}{\sqrt{[|SW_{xy}(a, \tau)| \cdot |SW_{yy}(a, \tau)|]}} \tag{5}$$

Consistency is defined as being the modulus of the crossed spectrum, normalized of the same spectrum, having values between zero and one, and represents the degree of linear between two processes. A value of 1 means a linear correlation between the two signals at a time *T* on the scale *a* and a value of 0 that indicates a zero correlation [34,39].

This method therefore makes it possible, in our case, to be able to assess and describe the links existing between rainfall variability in the Sahel and climatic (represented by the AMO, TNA, and TSA indices), both for the different scales (frequencies) and according to the evolution of relationships over time.

The statistical significance of the fluctuations observed by the wavelet transform is evaluated by comparing the local wavelet spectra against randomly distributed theoretical spectra. More details on these statistical tests are indicated and discussed in the associated literature [38,39].

In this article we consider that the percentages of consistency express a weak relationship when this percentage is less than 50%. The relationship is average when the percentages are greater than 50% and less than 70%. The relationship becomes strong when the percentages are between 70% and 90% and very strong if the percentage exceeds 90%.

## 3. Results

### 3.1. Sahelian Rainfall Trend

The results of statistical processing show an evolution marked by various cycles structured into three main periods (Figure 2).

The period of the fat cow, called as such in reference to the abundance of pastures, is visible between 1947 and 1969. This period is considered wet, as shown by the matrix carried out for the three countries. The number of rainfall-challenged years, with a negative index, is very small. Two years of deficit were observed in Mauritania (1948 and 1959) and in Burkina Faso (1947, 1959). Three years were observed for Senegal (1959, 1960, and 1968).

Starting with the 1970s, it was noted that a period of severe drought occurred throughout the entire Sahelian region (1970–1987—Mauritania; 1970–1990—Burkina Faso; and 1970–1998—Senegal):

In Mauritania, the succession of dry years was interrupted between 1970 and 1987. The drought was severe in 1977 and 1982–1983. During these years, the regional index exceeded −1.5.

For Burkina Faso, this long period is marked by a succession of years of deficit for two decades (except for 1974 [0.96] and 1976 [0.16]). Drought was severe during 1982, 1983, 1987, and 1997 when regional indices reached −2.08, −1.92, −1.44, and −1.44, respectively.

As for Senegal, the dry climatic phase extended over almost 30 years.

The period of the return of precipitations is marked by two clearly identified cycles.

The first period is characterized by a break in the dry conditions of the past (we observe an alternation of wet and dry years on the matrix). This trend may be interpreted as an intermediate period which started in Mauritania in 1988, in Burkina Faso in 1990, and in Senegal in 1999. This cycle, which overlaps the climatic conditions of the previous years, announces the major climate changes currently noticed in the West African region.

The second period is identified by a higher frequency of wet years (positive indices). As regards Mauritania, it was visible starting with 2003, with several wet years exceeding the +1 threshold (2003, 2005, and 2010). In Senegal, this trend was noted as early as 2008 with a succession of wet years that exceeded the index +1 (2008–2011), interrupted in 2011, which is considered to be a "normal" climatologic year. In Burkina Faso, the conditions are similar to those in Senegal. The last years of the pluviometric series (starting with 2008) marked the return to favorable conditions for rainfall. Except 2011, just like in Senegal, a negative index of −1.4 was recorded. The situation was the same in Mauritania as well, with an index of −0.96, which points to the aridity conditions of this Sahelian area in West Africa. This rainfall variability, which marked the entire region, was more pronounced in Mauritania, due to the geographical position of this country in relation to the West African Monsoon movement to the northern areas of Sahelian Africa.

We can note two normal years with a dryness trend in 2012 and 2013, for Mauritania, whereas throughout the rest of the area studied this interval is considered to be wet, with an index of +1.28 in Senegal and +0.80 in Burkina Faso. In 2013, rainfall quantities in these countries were relatively small, falling under moisture conditions.

Thus, after almost three decades of drought, a large part of Sahelian West Africa recorded a return of rains. This trend is subject to a differentiated spatial distribution. The intermediate period (the succession of wet and dry years) started in Senegal later. The wet cycle (higher frequency of wet years)

is less significant in Mauritania, even absent in the last years, namely 2012 and 2013 (this last year recorded a drought tendency). In 2014, there was a −1.57 index in Mauritania, whereas in Senegal and Burkina Faso there were wet conditions of +0.14 and +0.80, respectively.

*3.2. Rainfall Variability in Senegal and Burkina Faso*

The continuous wavelets transform is a good method to study the relation between rainfall and the climate index AMO, TNA, and TSA. This method was used by some authors to identify the non-stationary behavior of North Atlantic Oscillation (NAO) evolution. Similar researches were carried out by [34,40–46].

In this research, CWT was performed on monthly dataset aiming to identify the spectral components of the signals which can be assigned to different modes of variability characterizing the geophysical signal and, eventually, the time scales involved. The mean monthly values were available only for stations of Senegal and Burkina Faso used in the first part of this research (Rainfall, 1948–2017). Available monthly rainfall data were extracted between 1948 and 2017 for each country. These data vary from 697.27 mm for Senegal and 839.84 mm for Burkina Faso. We do not have monthly data for Mauritania and it is for this reason that this country is not associated with this analysis.

The CWT of the monthly data of precipitation was simulated (Figure 3). This analysis shows several variability scales for the two countries: Seasonal (6 months–1 year), interannual (2–3, 3–4, 4–6 and 5–7 years), quasi-decadal (QDO: 8–16), and multi-decadal (MDO: 18–24 only for Burkina Faso), as shown by Figure 3.

As for the annual band, it is of strong power for the two countries. The signal has no visible discontinuity. Seasonal variability is linked to the contrast between the dry seasons (November–March) and the wet seasons (April–October). Seasonal patterns normally weaken in arid years and only get stronger in wet years. This is not visible in Figure 3 and probably comes from the fact that we have used the monthly averages of the different stations in each country (a test carried out on the data of the Dakar station confirm this particularity).

The 2–3-year inter-annual band was present in both countries at specific dates. At the start of the series, we find it in 1950 for both countries. It was also present in 1969 in Senegal and in 1975 in Burkina Faso. In the 1990s, this band disappeared in Senegal and reappeared in the early 2010s, spreading to the 3–4 year mode. In Burkina Faso, this band reappeared between 1988 and 1995 then between 1998 and 2003 and finally, between 2008 and 2017.

The 3–4 year band is visible between 1950 and 1955 in Senegal and from 1948 to 1955 in Burkina Faso; it was also found between 1985 and 1995.

The 4–6 year band is present at the end of the series between 1995 and 2005 in Senegal and from 1995 until 2010.

The 5–7 year band was only present in Burkina Faso, between 1962 and 1972.

The quasi-decadal 8–16-year mode was present for the two countries studied. It was more spread out for Senegal but from 1975, it weakens and turns into 10–14-year-old fashion. In Burkina Faso, it was more powerful and concerns the period 1947–1985. There is a loss of energy after this last date.

The multi-decade mode does not appear on the wavelet of Senegal. It was present in Burkina Faso (18–24-years) from 1947 and until 1995 with a weakening since this last year.

## 4. Discussion

*Potential Relations between the Global Variability of AMO, TNA, TSA, and the Rainfall*

The West African Monsoon is a coupled atmosphere-ocean-land system and the major phenomenon of interest in the Sahelian zone in winter. Since the beginning of spring, temperatures increase, and a cold zone is formed in the Gulf of Guinea. This first thermal contrast explains the oceanic moisture transfer to the mainland in accordance with the trans-equatorial movement of trade winds in the southern hemisphere [34,41]. In West Africa, this transfer is governed by an overall movement of the

meteorological Equator and its low-pressure corridor (ITCZ, Intertropical Convergence Zone) towards the north, under the effect of the attraction of the Saharan thermal depressions and a greater vigor of the anticyclonic nuclei in the southern hemisphere in winter. Rainfall variability in the Sahelian area is a climate feature in this region. In the last years, real scientific progress has been made in understanding climatic mechanisms in this region and on large part of the planet as well [14,42]. Today, this knowledge allows one to state that climate is subject to natural fluctuations overlapped by some anthropic signals (global changes). The role of oceans in regulating convective flows has been extensively studied by several specialists [14,47–49]. Thus, due to studies on pressure fields and oceanic temperature, two natural signals known as; multi-decadal signal (a signal occurring over a period larger than 40 years—low frequency variability) and quasi-decadal signal (signal with a shorter period, of 8–14 years. low-frequency variability) have been identified [16,50–52].

According to these researches, the possible links between SST and rainfall conditions are very complex and should be investigated separately for each frequency by the use of high statistical methods as the wavelets. The correlation between the index climate AMO, TNA, TSA, and rainfall signal was computed by the coherence diagram for the different modes of variability. Wavelet consistency analyses identify significant common oscillations between two signals (precipitation/Atlantic TSM) at certain variability scales for certain time intervals [53]. The result of coherence wavelet with the global mode of variability (1, 2–4, 4–8, 8–16, 18–24, and 20–24-year are shown in the Figure 4 and Tables 3–5).

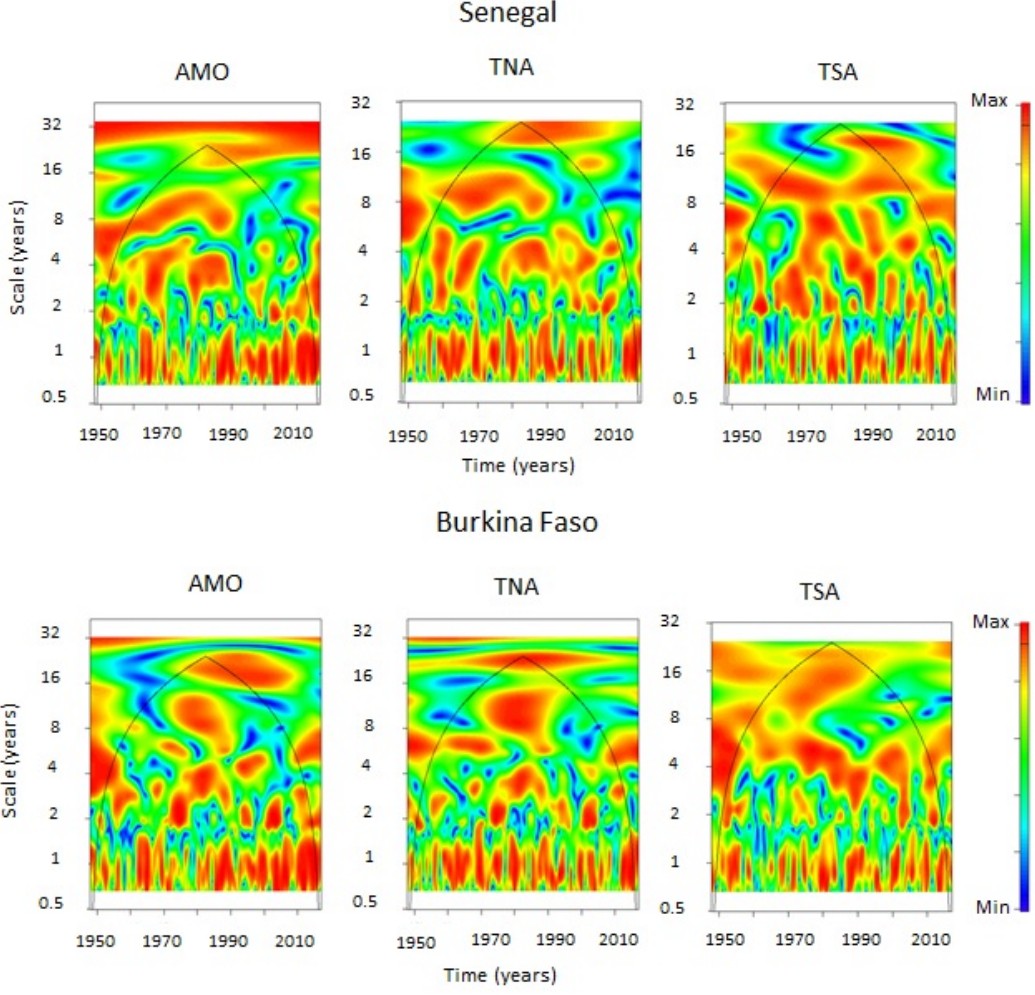

**Figure 4.** Wavelet coherence analysis between average monthly rain in Senegal and Burkina Faso (average of the different stations in each country) and climate indices: Atlantic multidecadal oscillation (AMO), Tropical Northern Atlantic Index (TNA), and Tropical Southern Atlantic Index (TSA).

The AMO shows moderate consistency with the monthly rainfall averages for the two countries studied.

On an interannual scale, the contribution of AMO is variable, we observe however a loss of coherence for the two countries between 1970 and 1980 (Table 3).

**Table 3.** Percentage of consistency between precipitation and the AMO index (%).

| Variability Mode | 1 | 2–4 | 4–8 | 8–12 | 8–16 | 18–24 | 20–24 |
|---|---|---|---|---|---|---|---|
| Senegal | 68.36 | 65.74 | 61.26 | 62.45 | 60.05 | 63.60 | 69.20 |
| Burkina Faso | 67.83 | 65.32 | 62.73 | 62.10 | 62.21 | 61.21 | 59.34 |

The contribution of the TNA on rainfall shows moderate consistency if we consider the different modes of variability studied (Table 4). The highest percentages are observed according to the modes of multi-decadal variability (20–24-years).

**Table 4.** Percentage of consistency between precipitation and the TNA index (%).

| Variability Mode | 1 | 2–4 | 4–8 | 8–12 | 8–16 | 18–24 | 20–24 |
|---|---|---|---|---|---|---|---|
| Senegal | 67.69 | 67.68 | 61.26 | 63.12 | 57.51 | 68.53 | 69.99 |
| Burkina Faso | 72 | 59.08 | 64.25 | 55.72 | 57.80 | 72 | 81.62 |

The consistency is well distributed in the multi-annual mode during the dry period for the two countries. In Burkina Faso, this strengthening was visible from the late 1960s to the mid-2000s, according to the 18–24 age group. For the 20–24 age group, this consistency is strong and reaches almost 82% (Table 4). Slightly offset in time, this reinforcement is also visible on the wavelet coherence spectrum of Senegal. We can see it between 1970 and 2005 but according to a lower consistency percentage of 72% for the 18–24-year mode (Table 4).

On a quasi-decadal scale (8–12 and 8–16 years), strong consistency is noted during the dry period. It is located between 1970 and 1990 for Burkina Faso with a loss of consistency before and after this date. In Senegal, this strengthening of coherence with TNA was visible between 1960 and 1985 and the loss of coherence appeared after this last date.

TNA's contribution to the interannual scale is variable. In Burkina Faso, the 4–8-year age group was reinforced between 1947 and 1975 then between 2004 and 2017. Between these two periods, a loss of power occurred. In Senegal, coherence is reinforced for this frequency band only between 1948 and 1952, there is an overall loss of power after this date. Despite the significant variability of the 2–4-year mode, we can nevertheless observe a structuring of this frequency band during the dry period. A strengthening was noted between the early 1970s and the mid-1990s for Senegal. In Burkina Faso, it was noted between the end of the 1970s and the mid-2000s. Reinforcement of consistency with the TNA was noted punctually according to different years for Burkina Faso (mid-1950s and 1960s) and for Senegal (early 1950s, mid 1960s, and early 2000s).

The consistency of the TNA on a 1-year interannual scale is very variable, however, there was a loss of power for the two countries in 2010.

The contribution of TSA to Sahelian rainfall is moderate according to the different modes of variability studied (Table 5). On a multiannual scale, this relationship is reinforced for the 18–24-year mode in Burkina Faso between 1947 and the mid-1990s and corresponds to a consistency of almost 79% for the 18–24-year mode. In Senegal, this link is not convincing, we note it by the loss of the signal observed on the wavelet graphs (Figure 4) but we find it quite intense according to the 16–20-year mode with 60. 14% of coherence between the late 1970s and early 2000s.

**Table 5.** Percentage of consistency between precipitation and the TSA index (%).

| Variability Mode | 1 | 2–4 | 4–8 | 8–12 | 8–16 | 18–24 | 20–24 |
|---|---|---|---|---|---|---|---|
| Sénégal | 66.16 | 67.79 | 69.70 | 75.82 | 57.51 | 53 | 48.51 |
| Burkina Faso | 63.62 | 63.81 | 73.84 | 55.72 | 67.21 | 78.91 | 75.24 |

On a quasi–decennial scale, the TSA contributes to the rainfall of Burkina Faso with a little more than 67% for the 8–16-year-old mode (Table 5). This frequency band is spread out over the entire period studied. However, we note that this relationship is more present during the dry period (from the late 1960s to the early 1990s). In Senegal the consistency of TSA is very low with this frequency band. However, there has been a strengthening of the 8–12-year-old mode (75.82%) (Table 5). This mode is generally present throughout the period studied.

Consistency with TSA on an interannual scale for mode 4–8 reached in Burkina Faso almost 74% (Table 5). It is reinforced at the start of the series and during the drought period (1970–2004) with a loss of power between 1985 and 1990 and a spread over the 2–4-year mode. In the 2–4-year mode, the consistency with TSA is close to 68% in Senegal and reaches 63.81% in Burkina Faso. In the latter country, this link is variable and is only reinforced at the start of the series, in the mid-1980s, in 2000 and in 2010. In Senegal, the consistency linked to this frequency is not well structured. However, some reinforcements occurred in the late 1960s, between 1960 and 1970, in the late 1970s, and late 1990 and early 2000s.

Consistency on an annual scale is very variable in the two countries and seems to present the same fluctuations.

A convincing link with the surface temperature of the oceans but difficult to define over time.

According to Caminade et al. [54] rainfall in the Sahel is characterized by a variability between 2 and 4 years in the Sahel, superimposed by slower oscillations (over 8–16 years) as well as a multi-decade evolution [55]. The results obtained thanks to wavelet coherence show convincing but different relationships according to the mode of variability and the geographic areas. We shall further summarize the main significant results of this analysis.

In Senegal AMO and TNA contribute more to rainfall according to the multiannual mode of variability (more than 69% for mode 20–24) while consistency with TSA is stronger according to the quasi–decennial and interannual modes (69.70% on the scale of 4–8 years and 75.82% for the mode 8–12 years).

For Burkina Faso, the contribution to the multi-annual scale of the TNA (81.62% for the 20–24-year mode) and the TSA (78.91% for the 18–24-year mode) is greater than that of the AMO (59.34% for the 20–24 mode). For the interannual and quasi-decennial scale the influence of TSA is stronger. It reaches 67.21% for the 8–16-year mode, 73.84% for the 4–8-year mode and 79.98% for the 4–6-year mode.

Mohino et al. [53] show that the climatic drought of the 1970s and 1980s corresponded to a negative multi-decadal oscillation of the Atlantic, favorable to a low rise in the convergence zone over Africa, whereas the 1950s and rainy 1960s, like the recent small rainfall recovery in the years 1990–2000, corresponded to a multi-decennial Atlantic oscillation returning to the positive phase. When studying the general rain trend, we have already highlighted the short periods of rainfall variability in the regions studied. The cycle of climatic drought started from the year 1970. This period extended until 1998 in Senegal but did not exceed the beginning of the 1990s for Senegal (Figure 2). The search for the impact of the surface temperature of the Atlantic Ocean over this period is illustrated in the graphs presented for the two countries (Figure 5). The influence of AMO seems intensified over this period for the 8–16-year mode for Senegal and Burkina Faso (even if for this last country we mentioned a weak global relationship above). For the same mode, we also highlighted a strong link with TNA during the dry period of the latter country (Figure 5). For the other modes this connection sometimes appears to be time-shifted or not clearly established.

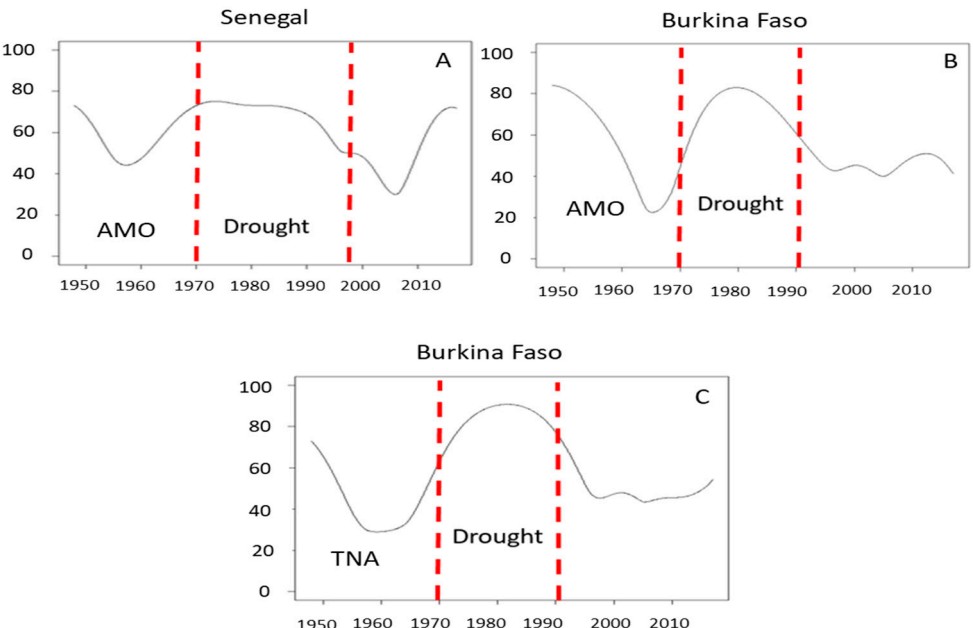

**Figure 5.** Percentage of consistency between precipitation and the AMO and TNA index for mode 8–16-years.

The connection between the drought period and ocean surface temperatures seems even more evident when one considers the percentage of dry years extracted from the graphic matrix (Figure 2).

Charts of Figure 6 show how the North Atlantic Ocean surface temperature (TNA) and the dry years' index (percentage of stations with a reduced centered index of cumulated precipitations lower than the fourth quintile in the studied area) vary. Variation curves show the reverse connection. Dry year indices are positive in 1970–1999, whereas the Atlantic Ocean surface temperature has negative values in 1971–1994 (the ocean was colder).

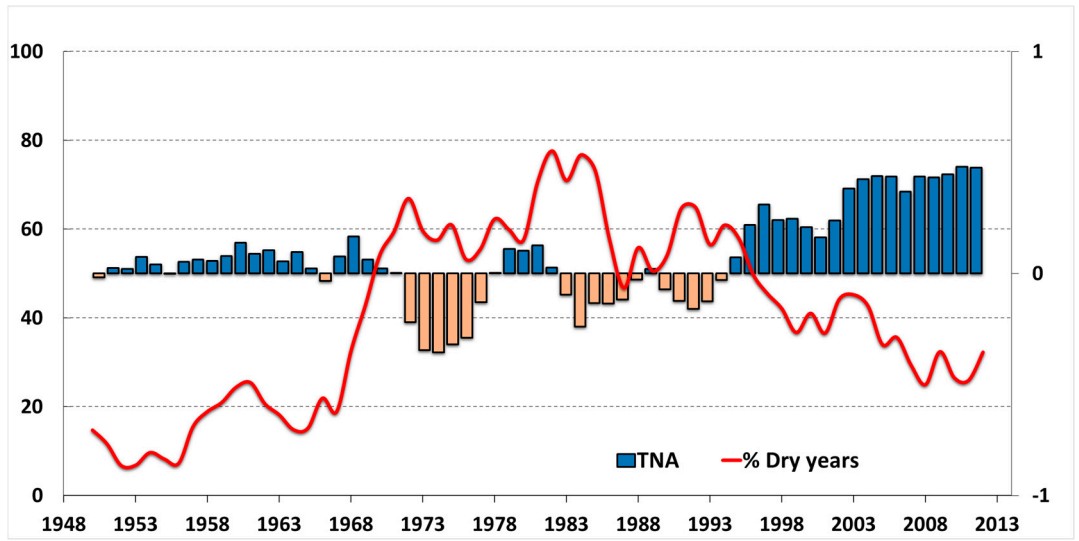

**Figure 6.** Evolution of dry years (%) and temperature of the Atlantic Ocean (TNA)—moving average over five years and the 1948–2014 interval; data source: https://www.esrl.noaa.gov/psd/data/correlation/tna.data.

In the last period of the series studied, the trend is reversed again, and, as the ocean surface temperature increases, the indices of dry years decrease in favor of rainier periods. Thus,

when quasi-decadal oscillations are positive and marked by a higher amplitude (warmer ocean), they correspond to a return of rains after Sahelian droughts and to an intensification of the West African monsoon rain cycle [54]. The graphical matrix (Figure 2) shows this return starting with 1990 in Burkina Faso, 1999-Senegal, and 2003-Mauritania. This explains the reverse connection with rainfall cycles in the studied regions.

## 5. Conclusions

The analysis of rainfall trend evolution shows that, following a long Sahelian drought, rains returned to this part of West Africa. The observations made in the entire Sahel region point to a major change that occurred in the mid-1990s (brutal alternation of dry and wet years) [54,55], which made some scientists use the term "ecologization" [56,57]. Debates on this issue are still underway within the scientific community, as there is some hesitation in evoking large-scale climate changes in the Sahel region. Therefore, regional contrasts have been clearly established, which is in line with the climate predictions established by the GIEC (2013) [37] for this area as well [48,54]. On a finer scale, the spatial organization described in numerous studies [58] shows a persistence of drought conditions in the extreme west of the Sahel as compared to eastern and central zones (a situation also encountered in the last years in other states such as Niger and Mali) [59,60].

The vulnerability of the area to these changes is, furthermore, acknowledged on a large scale, since much of its economy relies on the precipitation. Some specialists [61–64] have pointed out that the return of rains to the Sahel, even though they do not reach the wet cycle level of the 1950s, is directly related to the current level of greenhouse gases in the atmosphere. At the same time, these researchers minimize the impact of sea surface temperatures on local rainfalls. Another research shows, however, a greater importance of the influence of the ocean in the West African and Sahelian rainfall phenomena [14,65–67].

This connection with the ocean did not appear very strong according to the analysis that we have conducted on the monthly precipitation recorded in Senegal and Burkina Faso. Thus, we have highlighted the strongest coherences on a multi-annual scale with TNA and on an inter-annual scale for TSA in Burkina Faso. In Senegal, on the other hand, the influence of the AMO seems stronger on the pluviometry of this country, for the multi-annual and quasi-decennial modes. Finally, the relationship with the Sahelian drought seems more evident in the quasi-decadal mode (8–16 years). This connection is also very well illustrated in Figure 6.

**Author Contributions:** Each author has a half of contribution. Both authors have the same degree of contribution in terms of research concepts and methodologies. Z.N. analyzes the statistical-mathematical data strings by the Bertin type matrix method. It also has contributions in the comparative analysis and validation of AMO. TNA. TSA data. O.M. contributed to the article also through concept and research methodology. The editing and translation of this material have been reviewed and completed by O.M. Also, authors contributed to making the changes required by the reviewers. All authors have read and agreed to the published version of the manuscript.

**Funding:** The payment for the publication was made yesterday, 17.06.2020, by the author of the correspondence. The bank transfer was made, around 9.30 am, from Banca Transilvania to Credit Suisse (Switzerland). INVOICE ID water-817749 was passed on the transfer order.

**Conflicts of Interest:** The authors declare no conflict of interest.

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
