# Peer review of "Rainfall Variability and Trend Analysis of Rainfall in West Africa (Senegal, Mauritania, Burkina Faso)"

_water, doi:10.3390/w12061754_

Round 1

Reviewer 1 Report

Overall comments: This manuscript is an analysis of Sahelian rainfall variability with a discussion of correlations with Atlantic ocean temperatures. The manuscript is much better than the first version that I reviewed a few months ago. The methodology section is much improved. Their figures were also much improved and the inclusion of the wavelet analysis is a step in the right direction. However, I have one major issue below that needs addressed before I can recommend publication. Decision: Major Revisions.

Major comments:

  • The wavelet coherence analysis and percentages of consistency was interesting but considering I have no experience with the technique I have no way to understanding its statistical significance. You talk about numbers above 70% being “good” and numbers less than 60% as “bad” but you need to express this in terms of the literature. Is there a certain percentage of consistency that can be seen as statistically significant? This section needs more clarification.

 Minor comments:

Page 1, 1st paragraph: Use a more specific term than air humidity.

Page 1, bottom paragraph: The rainy season of June-September/October is summer, not winter.

Page 2, end of Introduction: Can you put this into perspective? What is the scale of the HDI?

Page 2, section 2: The introduction mentions the AMO, but you never mention anything about the TNA and TSA. I do a lot of work with teleconnections and I am not at all familiar with them. You need to have a brief discussion about what the TNA and TSA represent physically and how they affect African climate.

Page 5, last sentence: Should be -2.08

Page 7, 1st paragraph of section 3.2: You need to spell out NAO here since it is the first use.

Page 9, section 4.1: I wouldn’t call a periodicity greater than 40 years “high frequency variability”. Both quasi-decadal and multidecadal variability represent different modes of low-frequency variability.

Figure 4 caption should be in English.

Author Response

Major comments:

  • The wavelet coherence analysis and percentages of consistency was interesting but considering I have no experience with the technique I have no way to understanding its statistical significance. You talk about numbers above 70% being “good” and numbers less than 60% as “bad” but you need to express this in terms of the literature. Is there a certain percentage of consistency that can be seen as statistically significant? This section needs more clarification.
  • In this article we consider that the percentages of consistency express a weak relationship when this percentage is less than 50%. The relationship is average when the percentages are greater than 50% and less than 70%. The relationship becomes strong when the percentages are between 70 and 90% and very strong if the percentage exceeds 90%.

 Minor comments:

Page 1, 1st paragraph: Use a more specific term than air humidity.

We used water vapours instead of air humidity

Page 1, bottom paragraph: The rainy season of June-September/October is summer, not winter.

We changed the term with rainy season

Page 2, end of Introduction: Can you put this into perspective? What is the scale of the HDI?

We added the sentence: HDI, this index varies from the highest value 0.954 for Norway to the lowest value 0.354 for Niger

Page 2, section 2: The introduction mentions the AMO, but you never mention anything about the TNA and TSA. I do a lot of work with teleconnections and I am not at all familiar with them. You need to have a brief discussion about what the TNA and TSA represent physically and how they affect African climate.

We explainded AMO, TNA,TSA in discussion sector, figure 4

Page 5, last sentence: Should be -2.08

We changed with -2.08 and we insert the paragraph at page 6

Page 7, 1st paragraph of section 3.2: You need to spell out NAO here since it is the first use.

We spell-it North Atlantic Oscilation

Page 9, section 4.1: I wouldn’t call a periodicity greater than 40 years “high frequency variability”. Both quasi-decadal and multidecadal variability represent different modes of low-frequency variability.

We replaced:

Thus, due to studies on pressure fields and oceanic temperature, two natural signals known as multi-decadal signal (a signal occurring over a period larger than 40 years – low frequency variability) and quasi-decadal signal (signal with a shorter period, of 8-14 years, low-frequency variability) have been identified [55, 56, 57, 58] (pp. 602-607, pp. 1027-1030, pp. 17891-17896, pp. 1-4 ).

Figure 4 caption should be in English.

We changed in english

Reviewer 2 Report

Review of the manuscript entitled:  Rainfall Variability and Trend Analysis of Rainfall in  West Africa (Senegal, Mauritania, Burkina Faso)

                                          by

Zeineddine Nouaceur  and  Ovidiu Murarescu

The project regards the identification of rainfall variability in West Africa Sahel. This is  characterized by a long dry  season and a rainy season starting from  June to September-October. Such a  rainy season is due to the oceanic moisture transfer to the mainland that in turn  is governed by the Intertropical Convergence Zone moving towards the north.

The authors  analyse  data from 27 Sahelian weather stations located in three  different countries (Burkina Faso, Mauritania and Senegal) relative to  1947 - 2014 interval. 

The authors compare the rainfall variability with three regionalized indices of surface temperatures:  AMO, TNA,TSA. To do this, they use the   classical  wavelet method together with  a simple and qualitative   method called MGCTI .

The authors use a regional index (RI) to characterize the droughts .

They note that, starting from  1970s,   a period of severe drought occurred throughout the entire Sahelian region (1970-1987  in  Mauritania; 1970-1990  in  Burkina Faso; 1970-1998  in Senegal). During these years, RI exceed -1.5.  Drought was severe during 1982, 1983, 1987 and 1997 when RI  reached -2.08, -1.92, -1.44, - 1.44.

Very interesting are the results about the inverse  relationship between TNA and the dry years’ index.  Dry year indices are positive in 1970-1999, whereas TNA has negative values in 1971-1994 owing to  a colder ocean. As  TNA  increases, the indices of dry years decrease to favour  rainier periods. When quasi-decadal oscillations are positive and marked by a higher amplitude (warmer ocean), they correspond to a return of rains after Sahelian  droughts and to an intensification of the West African monsoon rainy cycle.

In general, I find the paper a good  contribution to the characterization of drought  over the sahel region .

Critical point:

The authors must explain how  the  yearly rainfall values (interval 1947-2014) were calculated within the  Burkina Faso, Mauritania and Senegal countries.  If the data are computed according to the used  interpolation process, I am skeptical on their availability. This because  the   only way to get climatic information is to make measurements and to enlarge the available meteorological network.

The English must be corrected by a native speaker

The figures must be more clear

 Anyway,  in my opinion this kind of papers is important, because, beyond the above specific aspects of network weakeness, offers a first way to   obtain a rapid climatological information waiting for  a denser meteorological network (see:   Lovejoy, Schertzer, Ladoy,.,  Fractal characterization of inhomogeneous geophysical measuring networks. Nature, 319, 43±44, 1986).

The authors are invited to investigate possible external forcing of the ascertained drought indices  like:

1)Solar cycles  (see: Reddy, Nerolla, Godson:The solar cycle and Indian rainfall, Theor. Appl, Climatol, 39, 194-198, 1989).

2)Influence of El Nino phenomenon  (see: Mazzarella, Giuliacci, Liritzis : On the 60-month cycle of Multivariate ENSO Index, Theor. Appl. Climatol.,  100, 23-27,  2010.

Author Response

Critical point:

The authors must explain how  the  yearly rainfall values (interval 1947-2014) were calculated within the  Burkina Faso, Mauritania and Senegal countries.  If the data are computed according to the used  interpolation process, I am skeptical on their availability. This because  the   only way to get climatic information is to make measurements and to enlarge the available meteorological network.

The English must be corrected by a native speaker

The article was reviewed by a Lecturer in english Language and Literature (see the atachment) 

The figures must be more clear

All the figures was reviewed

 Anyway,  in my opinion this kind of papers is important, because, beyond the above specific aspects of network weakeness, offers a first way to   obtain a rapid climatological information waiting for  a denser meteorological network (see:   Lovejoy, Schertzer, Ladoy,.,  Fractal characterization of inhomogeneous geophysical measuring networks. Nature, 319, 43±44, 1986).

The authors are invited to investigate possible external forcing of the ascertained drought indices  like:

1)Solar cycles  (see: Reddy, Nerolla, Godson:The solar cycle and Indian rainfall, Theor. Appl, Climatol, 39, 194-198, 1989).

2)Influence of El Nino phenomenon  (see: Maz

zarella, Giuliacci, Liritzis : On the 60-month cycle of Multivariate ENSO Index, Theor. Appl. Climatol.,  100, 23-27,  2010.

We improved the references list

Round 2

Reviewer 1 Report

This is a much improved manuscript compared to previous versions. All of my initial recommendations have been followed. 

Reviewer 2 Report

The paper has been revised  according my suggestions and now can be published 

This manuscript is a resubmission of an earlier submission. The following is a list of the peer review reports and author responses from that submission.

Round 1

Reviewer 1 Report

The manuscript by Nouaceur and Murarescu is about rainfall variability from 1947-2014 to 3 west African countries, Senegal, Mauritania and Burkina Faso. West Africa is a region that is expected to face great challenges in terms of future precipitation and temperature change and thus great caution is needed in that particular region.

After reading the manuscript  I feel that there are many obscure major points that in my opinion prevent the manuscript for publishing in its current form.

The main issues I have identified are:

  • The authors use station data from 3 countries that are far away from each other, not in a continuous domain, and yet they averaging all stations to define margins for their methodology. This should be further detailed and justified, because as it is right now in its current form is in my opinion scientifically not correct.
  • Introduction needs enrichment, why are the authors doing what they are doing, what is the situation currently there, who is affected from their study, refer to similar studies, etc.
  • Methodology is very limited. Authors should clearly define in different sections their methodological approaches thoroughly and comprehensively. For example they are using 2 approaches, however these should be divided into sections where each approach is elaborated separately. Also this issue with averaging all stations should not be ignored. Also trend analysis is not clear how it is happening. The authors should make a different section in the methodology that explains how trend analysis is being performed, and they should consider implementing metrics such as Mann-Kendall trend analysis or similar that have extensively used in similar studies.
  • Results are limited and figure 2 which is the main figure should be clearer and easier to read. I suggest the authors to make individual figures per country that are easier to read.
  • Discussion contains parts that should be in the results section. It should be re-written in a way that relates to the findings, not to present more results. Discussion should reflect on the findings.
  • All sections are in great need of enrichment. I made specific comments on the attached PDF.
  • Another issue is that precipitation is not only temporal, but has a spatial domain as well. This is of extreme importance, especially in West Africa where local rainfall variability is quite high. Authors make no mention of that, which cannot be neglected in my opinion.

Reviewer 2 Report

The subject addressed in the paper is very interesting but the authors commit errors of interpretation. I give some ideas to redirect the work.

As suggested by the authors, the rainfall oscillation highlighted in Figure 2 results from sea surface temperature anomalies, but the method used is far too vague. Clearly the oscillation results from the 64-year average period SST anomaly along the subtropical gyre that is in phase with the rainfall oscillation (see Figure 2c, d, J. Mar. Sci. Eng. 2018, 6, 107; doi:10.3390/jmse6030107).

So Figures 3 and 4 are not relevant. Accumulated energy of tropical cyclones probably results from the SST anomaly. Either way this does nothing in context unless the authors prove otherwise.

Regarding the possible anthropogenic impact, it could appear in the resumption of rainfall in the 2000s which seems rather erratic.

On the other hand, the authors should pay more attention to the presentation, explain the meaning of Figure 2 (what does x-axis represent?) make the labels intelligible.

I can only reject the paper with regret, encouraging the authors to resubmit it after they have integrated the latest developments in physical oceanography.

Reviewer 3 Report

Overall comments: This manuscript is an analysis of Sahelian rainfall variability with a discussion of correlations with Atlantic ocean temperatures. The manuscript is very well done. The authors did a nice job of explaining their methodology and objective. I really like the idea of using the color scheme to show temporal and spatial variability of precipitation. Their figures were also informative. I did have two major issues and several minor issues that are detailed below. I don’t think any of my issues are deal-breakers for the manuscript, but in sum they push me towards a decision of Major Revisions.

Major comments:

My biggest points of contention with this otherwise excellent manuscript is 1) The unnecessary inclusion of figure 4 and 2) the lack of statistical analysis for figure 3.

  • I have a problem with how you discuss figure 4. You show the relationship between dry years and the Sahel and North Atlantic ACE. Whereas you can make a strong meteorological argument that the ocean temperatures discussed in figure 3 are driving variation in Sahel precipitation, you cannot do the same for ACE and Sahel precipitation. Instead, the literature shows that ACE is also being driven by Atlantic Ocean temperatures. The proper thing to do is to correlate TNA with ACE, but since this manuscript is not about ACE, there is no reason to include ACE at all. So Figure 4 and all discussion about ACE should be deleted.

  • Figure 3 is a compelling image, but more statistical information is needed. You should do some kind of correlation between TNA and your % dry years to show that the relationship is indeed statistically significant. If you use the 5-yr moving average of % dry years you will have to consider the reduced degrees of freedom when you determine statistical significance. Alternatively, you can simply correlate TNA with annual % dry years to eliminate the need to reduce degrees of freedom.

Minor comments:

Line 30: Use a more specific term than air humidity.

Line 59-61: Can you put this into perspective? What is the scale of the HDI?

Line 92: I didn’t understand the method of constructing the index. How exactly did you reduce and center the data for the stations? Can you provide an example? And I also don’t understand the scale. Is something missing? It goes from + infinity to -1.80 infinity? How do you get -2.08 in 1983 for Burkina Faso? I calculate -1.85 (6 stations were -2 and one was -1 then divide by 7 stations). From what you describe, the limits should be plus or minus 2.0, like what you show for 1992 in Senegal where all 7 stations were very dry (-2 * 7)/7 = -2.0. I would double check the math on all of these indexes. Most look correct, but some look wrong.

Figure 2: What is the dashed line? Some kind of moving average?

Line 137: You mean -0.96.

Figures 3 and 4: Make sure your legend is in English.